# Health care expenditure in the last five years of life is driven by morbidity, not age: A national study of spending trajectories in Danish decedents over age 65

Anne Vinkel Hansen[1,2]*, Laust Hvas Mortensen[1,2], Stella Trompet[3], Rudi Westendorp[1,2]

**1** Data Science Lab, Statistics Denmark, Copenhagen, Denmark, **2** Section of Epidemiology, Department of Public Health, University of Copenhagen, Copenhagen, Denmark, **3** Section of Gerontology and Geriatrics, Department of Internal Medicine, Leiden University Medical Center, Leiden, The Netherlands

* aih@dst.dk

## Abstract

### Background

The high level of medical spending at the end of life is well-documented, but whether there is any real potential for cost reductions there is still in question, and studies have tended to overlook the costs of care.

### Aim

To identify the most common health care spending trajectories over the last five years of life among older Danes, as well as the determinants of following a given trajectory.

### Methods

We linked Danish health registries to obtain data on all health care expenditure (including hospital treatment, prescription drugs, primary care and costs of communal care) over the last five years of life for all Danish decedents above age 65 in the period 2013 through 2017. A latent class analysis identified the most common cost trajectories, which were then related to socio-economical characteristics and health status at five years before death.

### Results

Total health care expenditures in the last five years of life were largely independent of age and cause of death. Costs of home care and residential care increased steeply with age at death whereas hospital costs decreased correspondingly. We found four main spending trajectories among decedents: 3 percent followed a late-rise trajectory, 11 percent had accelerating costs, and two groups of 43 percent each followed moderately or consistently high trajectories. The main predictor of total expenditure was the number of chronic diseases.

**Data Availability Statement:** Due to restrictions in Danish law, the confidential health care data used in this study can only be accessed through

Statistics Denmark, the state organization holding the rights to the data. Danish scientific organizations can be authorized to work with data within Statistics Denmark and can provide access to individual scientists inside and outside of Denmark. Data are available via the Research Service Department at Statistics Denmark: (www.dst.dk/da/TilSalg/Forskningsservice) for researchers who meet the criteria for access to confidential data. The authors of this study had no special access privileges others would not have.

**Funding:** This study was funded by an unconditional research grant from Novo Nordisk Fonden (NNF17OC0027812). AIH and LHM are employed at Statistics Denmark, the national Danish Statistics office, which organization holds the rights to the registry data used in the study. Otherwise, the funders had no role in study design, data collection and analysis, decision to publish, or preparation of the manuscript.

**Competing interests:** AIH and LHM are employed at Statistics Denmark, the national Danish Statistics office. This does not alter our adherence to PLOS ONE policies on sharing data and materials.

## Interpretation

Spending at the end of life is largely determined by chronic disease, and age and cause of death only determine the distribution of expenses into care and cure.

## Introduction

Worldwide, societies struggle to deliver quality healthcare amidst challenges posed by population ageing and the emergence of age-associated morbidity. One area of attention is the relatively high proportion of health care expenditure that occurs towards the end of life [1, 2]. The high cost of dying has been suggested to be the driver of increasing health care expenditures in ageing populations [3], although the full picture [2, 4, 5] seems to be that increases in hospital costs with age are driven by a combination of age and proximity to death. Interestingly, several studies [6–9] have shown that hospital expenditures over the last year of life actually decrease with age, which has been taken to indicate rationing of healthcare and possible undertreatment of the elderly.

Towards the end of life, the population becomes more heterogeneous with diverging patterns of functional decline and morbidity [10] which are reflected in dissimilar trajectories of health services needs. Studies of [11–13] the distribution of these trajectories can potentially allow us to detect the drivers of health care expenditures and discover sub-populations for which interventions may reduce health care related costs. Some such studies aim to identify groups [12] that are potential candidates for switching to palliative treatment, a move which could potentially improve patient satisfaction while simultaneously decreasing hospital costs. There is, however, some indication [1, 11] that health care expenditures at advanced ages are driven by the presence of multimorbidity, in which case the potential for end-of-life interventions reducing health care expenditure at the end of life is limited.

This paper examines the distribution of health care expenditures over the last five years of life for Danish decedents aged 65 years and over. The health care systems of the Nordic countries share basic features with the National Health Service (NHS) models in the United Kingdom, New Zealand, and southern Europe [14]. The funding is tax based, most services are free of charge, and the main actors, the purchasers and most of the suppliers are public. Importantly for this research, detailed individualized information on health care expenditures is available via population registers at the office of national statistics, Statistics Denmark. Studies on health care expenditure at the end of life often make opportunistic use of health insurance data or hospital records, and thus are liable to miss significant expenses not relevant to their data sources. We analyze total health care including hospital costs, primary health care, prescription drugs, and communal care. We show the distribution of expenditures by age, sex, calendar year, and cause of death, and examine the distribution of the individual trajectories over the last five years of life in order to look for determinants of high- and low-cost trajectories.

## Methods

### Study population

The study population consisted of all individuals dying at age 65 years or older in Denmark in the period 2013–2017. We excluded those who had not been resident in Denmark for at least five years before date of death. Primary ICD-10 codes for cause of death were obtained from the register of causes of death [15] and grouped into main categories, using the A-list

published by the Danish National Health Data Authority [16] and splitting dementia out as a separate category. For more details, see S1 Table.

## Health care expenditure

The last five years of life were divided into three-month intervals and per-person health care expenditures computed for each time interval, in total and by type of expenditure; i.e. hospital costs, primary care, prescription drugs, home care, and residential care. Expenditures were deflated to 2010 price levels [17].

**Treatment-related costs: Hospital, primary care and prescription drugs.** For hospital costs, the data used was the DRG-grouped National Patient Register [18] which has information on all in- and out-patient somatic and psychiatric hospital contacts in Denmark. The prices assigned for Diagnosis Related Grouping (DRG–for inpatients) and the corresponding Danish Ambulant Grouping System (DAGS–for outpatients) are average cost rates assigned to a treatment type. Costs for hospital admissions stretching across more than one three-month period were divided proportionally into the time intervals.

The source for primary care is the National Health Insurance Service Registry [19], which collects data on all treatment covered by national health insurance as provided outside of a hospital setting. Costs for prescription medicine sold by pharmacies was sourced from the Danish National Prescription Registry [20].

**Care-related costs: Home care and residential care.** For home care, we used registry data from Statistics Denmark [21], containing individual-level information on the number of hours of home care provided by the Danish municipalities. We estimate the costs of home care from a database of reimbursement rates for private providers of home care [22]. As for residential care, Statistics Denmark has data for the period 2008–2016 on all individuals living either in nursing homes or residential homes for the elderly [23]. We extended these data to 2017 by assuming that individuals living on addresses that were nursing/residential homes in 2016 were living in nursing/residential homes. We were thus able to estimate the number of days spent in nursing homes for each individual. From 2018, Statistics Denmark has data on the total sum spent by each municipality on home care and residential care respectively. We used these data, combined with data on hours on home care delivered and days spent in residential care to estimate how the price of one day in residential care compared to the price of one hour of home care. Assuming this number to be constant across the period, we can estimate the cost of one day of residential care using the data on costs of home care described above.

## Statistical analyses

**Identification of the most common spending trajectories.** We used group-based trajectory modeling [24, 25] to identify the most common patterns of healthcare expenditure over the last five years of life. This method groups the population into data-derived classes of individuals following similar trajectories. To select the number of classes and shapes of the trajectories, we tested a range of models on a ten percent random subsample of the data, describing the average cost trajectories as first to third degree polynomials with dummy variables for the last quarter of life, and testing models with one to eight latent classes. We chose a final and fitted this model to the entire dataset. For these analyses, we modeled log-transformed quarterly expenditures as normal. We fit the models in R using the lcmm package [26].

**Modeling approach.** We originally planned to select the final model based on BIC. However, when running the analyses, BIC kept decreasing past a point where the models became less clinically interpretable. When the trajectory analysis is seen more as a way of

approximating a complex underlying distribution of trajectories than a detection of true underlying groups, this is an expected phenomenon in large samples, and we settled on a four-class model as a reasonably complex yet interpretable choice. Of the three four-class models tested, the one using quadratic polynomials had the lowest BIC, but the models were very similar. For more details on the modelling process, see S1 Appendix.

The choice of a model which didn't model within-class variation over one that did was partly due to the computational intensity of the latter models, partly due to a preference for the Latent Class Trajectory Modeling framework of thinking of the latent trajectories as approximations of an underlying distribution of trajectories, rather than as discrete, "real", underlying groups [27]. This reasoning also led us to model residual variances and the variance-covariance matrix as fixed across classes.

The estimation algorithms used are vulnerable to local optima and the choice of starting values can affect the outcome massively. For reasons of computational time, we used the LCMM package option of starting parameter search at the estimates of the one-class solution. We reran the final model using multiple starting points and found no changes.

**Predictors of latent class membership.** Potential predictors for class membership studied were age at death, immigrant status, an affluence index [28] combining income and savings at age 65 and assigning a cohort-adjusted percentile to each individual, marital status, region of residence at time of death, cause of death, the number of chronic diseases, and presence of a list of individual chronic disease groups: Cancer, heart disease, other cardiovascular disease, respiratory disease, dementia, diabetes, asthma/allergy, and mental illness/epilepsy.

## Results

In the period 2013–2017, we identified 218,231 decedents aged 65 years and over, after having excluded 602 due to the person not having been resident in Denmark for the last five years of life, and 174 due to the person being missing from population registries. The mean age at death was 83.6 (IQR 77–91) for women and 79.9 (IQR 73–86) for men.

### Health care expenditure by age, sex, year, and cause of death

The total mean health care expenditures over the last five years of life was 85.8 thousand euro per person but the distribution was highly skewed, with the top five percent of decedents accounting for 18% of the total expenditures. Table 1 presents the distribution of per-person health care expenditures by cause of death, age, sex, and year of death. The total healthcare expenditures were largely independent of age at death. The main contributors to the total expenditures were hospital costs (56% of total), home care (21%), and residential care (15%) but these categories were unevenly distributed according the cause of death. Deaths from cancer had the highest mean hospital costs and dementia the lowest (65 and 22 thousand euro respectively) whereas expenses were the opposite for communal care (12 and 64 respectively).

Total mean health care expenses were consistently higher in women than men for all causes of death, with the difference ranging from one thousand euro for cancer to eight thousand euro for heart disease and dementia. There was no clear association between total health care expenditure and the age of death, and no association with calendar time (after correcting for inflation).

Whereas the total healthcare expenditures were largely independent of age at death, the distribution of the costs was markedly different (Fig 1). Hospital costs decreased from a mean of 70 thousand euro in the decedents aged 65–74 to a mean of 22 thousand euro in those aged 95 +, while home care and residential care increased from means 9 and 4 thousand euro to means 34 and 29 thousand euro. Costs of prescription drugs and primary care over the last five years

**Table 1. Mean healthcare costs (per-person 1000 EUR) and inter-quartile range (IQR) across the last five years of life by type of expenditure, sex, age and cause of death.**

| . | | All | | Cancer | | Heart disease | | Other diseases of circulatory system | | Diseases of respiratory system | | Dementia | | Any other | |
|---|---|---|---|---|---|---|---|---|---|---|---|---|---|---|---|
| | | mean | (IQR) | mean | (IQR) | mean | (IQR) | mean | (IQR) | mean | (IQR) | mean | (IQR) | mean | (IQR) |
| **Type** | | | | | | | | | | | | | | | |
| | Total | 86 | (37–111) | 83 | (39–108) | 75 | (27–102) | 85 | (34–111) | 94 | (43–122) | 92 | (60–111) | 91 | (34–118) |
| | Hospital | 48 | (14–64) | 65 | (27–86) | 40 | (10–53) | 40 | (13–53) | 50 | (17–66) | 22 | (5–29) | 46 | (11–58) |
| | Home care | 18 | (0–14) | 8 | (0–4) | 16 | (0–15) | 23 | (0–18) | 22 | (0–22) | 25 | (0–28) | 23 | (0–21) |
| | Residential care | 13 | (0–16) | 4 | (0–0) | 11 | (0–8) | 15 | (0–26) | 12 | (0–16) | 39 | (14–61) | 14 | (0–21) |
| | Drugs | 5 | (1–6) | 4 | (1–5) | 4 | (2–6) | 4 | (1–5) | 7 | (3–10) | 5 | (2–6) | 5 | (1–6) |
| | Primary care | 3 | (1–3) | 2 | (1–3) | 3 | (1–3) | 3 | (1–3) | 3 | (1–3) | 2 | (1–3) | 3 | (1–3) |
| **Sex** | | | | | | | | | | | | | | | |
| | Female | 88 | (41–114) | 83 | (40–109) | 79 | (31–106) | 87 | (38–112) | 97 | (47–125) | 95 | (63–112) | 93 | (40–119) |
| | Male | 83 | (34–109) | 82 | (39–107) | 71 | (23–97) | 84 | (30–109) | 90 | (40–118) | 87 | (53–109) | 88 | (28–115) |
| **Age** | | | | | | | | | | | | | | | |
| | 65–74 | 91 | (33–121) | 93 | (46–122) | 85 | (15–103) | 85 | (23–114) | 103 | (41–138) | 86 | (54–108) | 94 | (20–125) |
| | 75–84 | 85 | (37–111) | 79 | (39–103) | 73 | (24–100) | 84 | (32–111) | 95 | (45–124) | 88 | (53–109) | 95 | (35–125) |
| | 85–94 | 82 | (39–107) | 70 | (33–92) | 73 | (31–100) | 85 | (39–109) | 87 | (42–115) | 94 | (61–112) | 85 | (39–112) |
| | 95+ | 89 | (50–111) | 81 | (38–106) | 85 | (44–108) | 92 | (51–114) | 92 | (50–115) | 98 | (68–113) | 89 | (49–111) |

of life peaked when death occurred around age 80 years and decreased in those dying at older age.

## Spending trajectories

To further explore the heterogeneity in health care expenditures over the last five years of life, we used a trajectory model grouping the decedents into four latent classes (Fig 2). Few decedents (3%) followed a trajectory with low mean expenditures up until the very last year of life, and in this group, hospital costs made up 90% of the mean total of 13 thousand euro. A somewhat larger group (11%) followed a trajectory with accelerating mean expenditures, starting low but increasing steeply towards the end of life. The overwhelming majority of the decedents were in groups with persistently increasing mean expenditures, starting either from a moderate high (43%) or a persistently high costs level (43%). The mean total health care expenditure of 131 thousand euro for the High Persistent-Expenditures group was two- to tenfold higher than that of the other groups. The High Persistent Expenditures trajectory was characterized by both mean hospital costs and mean care costs starting high, increasing persistently over the last five years of life and spiking in the last quarter of life. This group had by far the highest mean proportion of care expenses at 38% (IQR 3–67%). The expenditures of the Moderate High group followed similar trajectories but had proportionally lower expenses and a lower mean proportion of care expenses (23%, IQR 0–43%).

## Sociodemographic determinants of spending trajectory

Table 2 describes the latent classes by sex, age, cause of death, socioeconomic variables and comorbidities. The Low and Accelerating Expenditures groups were younger (mean age 77.2 and 78.2 years respectively) and less likely to be female (41 percent women in both groups) than the Moderate Persistent and High Persistent Expenditures groups (mean ages 82.3 and 82.6 years, respectively 51 and 57 percent women). The socioeconomic variables did not vary

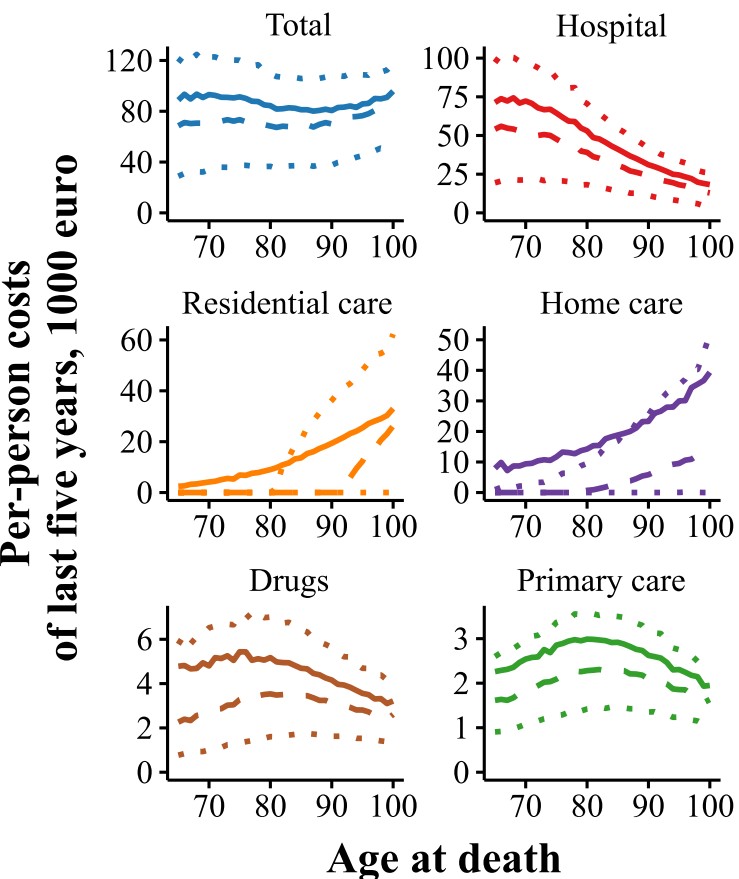

**Fig 1. Mean health care expenditure over the last five years of life by type of expenditure.** Mean (solid line), median (dashed) and quartiles (dotted).

much between the groups. The tendency for health care expenditure to shift from cure to care with age was still present within the groups, and there was systematic variation around the group mean total expenditure with age (S2 Table).

## Health-related determinants of spending trajectory

Death from cancer was particularly prevalent (43%) in the group of decedents with an Accelerating Expenditures trajectory but there were no other obvious associations with specific causes of death (Table 2). Whereas 85% of those in the Low Expenditures group suffered at most one chronic disease, 62% of the High Persistent Expenditures group suffered at least five comorbid diseases. The only chronic disease which did not increase in prevalence with the mean cost of the trajectory was cancer, which was most common in the Accelerating Expenditures trajectory.

## Discussion

For Danish decedents over the age of 65 dying in the years 2013–17, total health care expenditure across the last five years of life was largely independent of age and cause of death. However, the distribution by type of expenditure–hospital costs, home care, residential care, drugs,

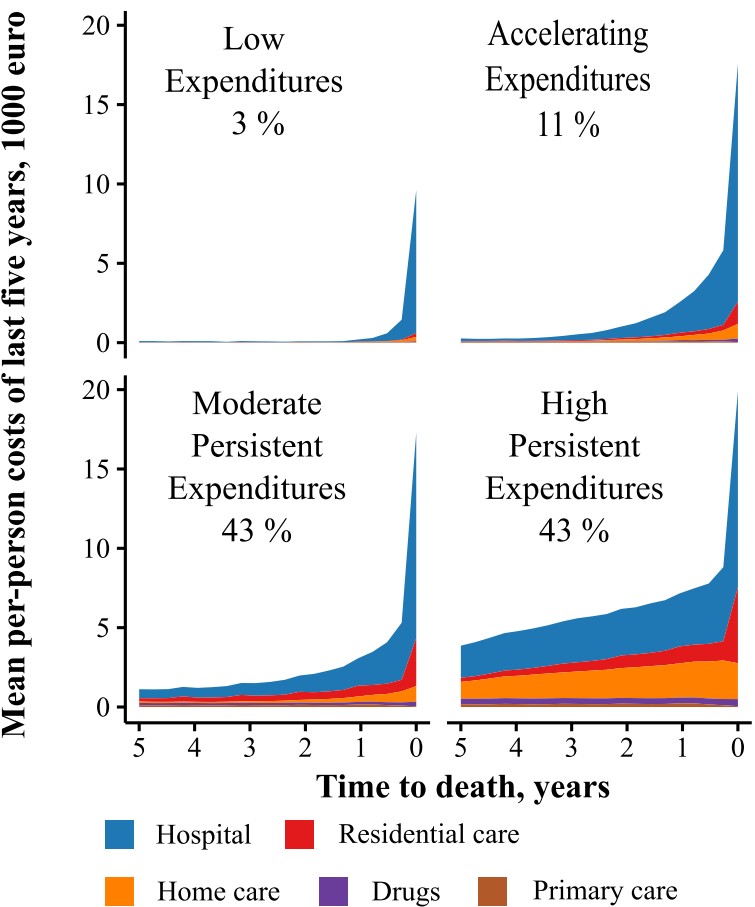

**Fig 2. Health care spending trajectories of decedents in the last five years of life.** Mean total costs in the Low-, Accelerating-, Moderate-Persistent, and High-Persistent-Expenditures groups were 13.4, 43.5, 56.9, and 131.3 thousand euro respectively.

and primary care–varied markedly dependent on age and cause of death. Hospital costs decreased dramatically with age, with reciprocal increases in costs for communal care service. Similarly, deaths from cancer had the highest mean hospital costs and the lowest mean care costs, while dementia deaths had the lowest hospital costs and the highest care costs.

The overall relative homogeneity in total health care expenditure covered over large differences in individual cost trajectories which were not adequately described by age and cause of death. A group-based trajectory analysis separated the decedents into four groups of similar trajectories. The strongest explanatory factor for the diversity in these spending trajectories was the number of chronic diseases. All groups had a characteristic spike in the last quarter of life, but only a small proportion of decedents had low expenditures up to the very last quarter. A larger group, possibly characterizable as a terminal-disease group, had rapidly accelerating costs toward the end of life. But the largest two groups, each comprising more than 40% of the population, had persistently increasing expenditures over the last five years of life, starting at a moderate or high level. We found two other studies [11, 12] which performed a similar data-driven detection of end-of-life trajectories based on healthcare expenditures. Our results somewhat resembled those of the study of Medicare decedents [11] with most of the population following persistently increasing trajectories, a progressive and a late-rise trajectory (although the

**Table 2. Description of the classes of the latent trajectory analysis.**

| | | Low Expenditures (n = 7,200) | | Accelerating Expenditures (n = 23,000) | | Moderate Persistent Expenditures (n = 94,830) | | High Persistent Expenditures (n = 93,201) | |
|---|---|---|---|---|---|---|---|---|---|
| Total health care expenditure over last five years of life (mean, IQR*) | | | | | | | | | |
| | | 13.4 | (0–18) | 43.5 | (15–62) | 56.9 | (29–77) | 313.3 | (81–154) |
| Proportion hospital costs (mean, IQR) | | | | | | | | | |
| | | 51% | (0–95) | 73% | (60–95) | 61% | (37–87) | 51% | (23–81) |
| Proportion care costs (mean, IQR) | | | | | | | | | |
| | | 4% | (0–0) | 13% | (0–12) | 23% | (0–43) | 38% | (3–67) |
| Total health care expenditure in last year of life (mean, IQR) | | | | | | | | | |
| | | 11.9 | (0–15) | 31.0 | (9–43) | 30.2 | (12–40) | 44.0 | (24–55) |
| Female (N, %) | | 2,939 | 40.8 | 9,453 | 41.1 | 48,702 | 51.4 | 53,131 | 57 |
| Age at death (N, %) | | | | | | | | | |
| | 65–74 | 3,284 | 45.6 | 9,001 | 39.1 | 21,200 | 22.4 | 19,512 | 20.9 |
| | 75–84 | 2,238 | 31.1 | 7,412 | 32.2 | 32,375 | 34.1 | 32,107 | 34.4 |
| | 85–94 | 1,438 | 20.0 | 5,665 | 24.6 | 34,251 | 36.1 | 34,150 | 36.6 |
| | 95+ | 240 | 3.3 | 922 | 4.0 | 7,004 | 7.4 | 7,431 | 8.0 |
| Marital status (N, %) | | | | | | | | | |
| | Married/Cohabiting | 2,636 | 36.6 | 10,755 | 46.8 | 37,919 | 40 | 32,243 | 34.6 |
| | Widow/er | 2,125 | 29.5 | 7,284 | 31.7 | 40,640 | 42.9 | 43,302 | 46.5 |
| | Divorced | 1,347 | 18.7 | 3,056 | 13.3 | 10,301 | 10.9 | 11,991 | 12.9 |
| | Never married | 1,092 | 15.2 | 1,905 | 8.3 | 5,970 | 6.3 | 5,665 | 6.1 |
| Immigrant (N, %) | | 456 | 6.3 | 964 | 4.2 | 3,183 | 3.4 | 3,345 | 3.6 |
| Affluence index (mean, IQR) | | | | | | | | | |
| | | 44.0 | (21–66) | 49.4 | (24–75) | 50.1 | (26–74) | 49.5 | (25–74) |
| Cause of death (N, %) | | | | | | | | | |
| | Cancer | 2017 | 28 | 9810 | 42.7 | 28179 | 29.7 | 21469 | 23 |
| | Heart disease | 1191 | 16.5 | 3356 | 14.6 | 17090 | 18 | 14757 | 15.8 |
| | Other circulatory | 675 | 9.4 | 2136 | 9.3 | 9437 | 10 | 8089 | 8.7 |
| | Respiratory | 569 | 7.9 | 1634 | 7.1 | 10041 | 10.6 | 15461 | 16.6 |
| | Dementia | 230 | 3.2 | 1017 | 4.4 | 8281 | 8.7 | 9094 | 9.8 |
| | Other | 2518 | 35 | 5047 | 21.9 | 21802 | 23 | 24331 | 26.1 |
| Number of chronic diseases (N, %) | | | | | | | | | |
| | 0–1 | 6155 | 85.5 | 10532 | 45.8 | 16965 | 17.9 | 6616 | 7.1 |
| | 2–4 | 964 | 13.4 | 10297 | 44.8 | 45812 | 48.3 | 28999 | 31.1 |
| | 5+ | 81 | 1.1 | 2171 | 9.4 | 32053 | 33.8 | 57586 | 61.8 |
| Prevalence of selected chronic diseases (N, %) | | | | | | | | | |
| | Cancer | 1546 | 21.5 | 10523 | 45.8 | 34110 | 36 | 31719 | 34 |
| | Heart disease | 1284 | 17.8 | 9193 | 40 | 56314 | 59.4 | 64656 | 69.4 |
| | Cardiovascular disease | 935 | 13 | 6283 | 27.3 | 32508 | 34.3 | 42130 | 45.2 |
| | Resporatory disease | 1239 | 17.2 | 8287 | 36 | 41999 | 44.3 | 57137 | 61.3 |
| | Dementia | 114 | 1.6 | 1797 | 7.8 | 13316 | 14 | 19841 | 21.3 |
| | Diabetes | 147 | 2 | 1375 | 6 | 14652 | 15.5 | 23143 | 24.8 |
| | Asthma | 37 | 0.5 | 283 | 1.2 | 3765 | 4 | 8670 | 9.3 |
| | Mental illness, epilepsy | 775 | 10.8 | 7347 | 31.9 | 45220 | 47.7 | 61224 | 65.7 |

* IQR: Inter-quartile range

late-rise group made up a larger proportion of Medicare decedents) and with the number of chronic conditions being the main factor explaining trajectory membership. We saw larger

age differences than the Medicare study, and unlike for Swiss decedents [12], we did not find the highest-cost latent classes to be younger but instead older than average. This is likely due to our study including care costs as well as hospital costs.

The main strength of this study is the availability of individual-level data for an entire population's use of health care services including care costs. As healthcare and eldercare in Denmarks is largely taxpayer-funded, the information available in registries accounts for 97% of personal health care expenditure in Denmark [14]. On the individual level, however, DRG rates are average rates and may not reflect the actual expenditures of each individual treatment, and since the computation of care expenditure involves a fair amount of estimation and imputation, there may be some misclassification. The proportion of expenditure on drugs is underestimated, as the cost of drugs used in a hospital setting is categorized under hospital costs. The groups of the latent class analysis should be thought of more as a way to decompose the variability of the population's trajectories than as "real" and essentially different underlying populations [27] and it should be emphasized that individual trajectories can vary a lot around the mean group trajectory and have different proportions of the various types of costs.

The relative independence of age in the population of decedents contrasts with a massive increase with age in mean health care expenditure for the total Danish population [5], which is expected as the high cost of dying and the increasing proportion of decedents with age drive mean individual health care expenditure up. We show that the decrease with age in end-of-life hospital costs observed both internationally [6–9] and in Denmark [2, 14], is only a part of the fuller picture where this apparent undertreatment [7, 8] of the elderly is made up by increasing communal care costs. This substitution of expenses from cure to care may well be a sign of appropriate personalized care in old age. Our study indicates that high spending at the end of life seems to largely reflect spending patterns set in motion much earlier in life and to be ultimately caused by multimorbidity–a phenomenon also observed by Davis et al [11], although for a shorter time horizon.

Why is multimorbidity so costly? The question is not as rhetorical as it might seem, and it informs the question of how to face the challenge of delivering healthcare to an ageing population. Multimorbidity increases the complexity of the individual's health, as well as the complexity of the system treating the individual. As individuals come closer to death, the number of parts of the organism affected by disease increase, as does the number of parts of the healthcare system involved. The question is to what extent the increase in healthcare costs with complexity of disease and with closeness to death is due to the biological complexity, and to what extent it is due to the complexity of the healthcare system. If we want to control healthcare costs in an ageing population, the key may be to improve the strategies for caring for older citizens with multiple chronic diseases.

## Supporting information

**S1 Fig. Trajectories of mean health care expenditure by cause of death.**
(EPS)

**S1 Table. Grouping of causes of death–definition and numbers.**
(DOCX)

**S2 Table. Healthcare expenditure (per-person 1000€) in the last five years of life, and proportion of hospital costs by age at death and latent class membership.**
(DOCX)

**S1 Appendix. Further description of the latent trajectory analysis.**
(DOCX)

## Acknowledgments

We thank Aksel Juel Clemmensen and professor Karsten Vrangbæk for feedback on the paper, and Jolien Cremers for the use of her algorithm to estimate costs of home care.

## Author Contributions

**Conceptualization:** Anne Vinkel Hansen, Laust Hvas Mortensen, Stella Trompet, Rudi Westendorp.

**Data curation:** Anne Vinkel Hansen.

**Formal analysis:** Anne Vinkel Hansen.

**Project administration:** Laust Hvas Mortensen.

**Software:** Anne Vinkel Hansen.

**Supervision:** Laust Hvas Mortensen, Stella Trompet, Rudi Westendorp.

**Visualization:** Anne Vinkel Hansen.

**Writing – original draft:** Anne Vinkel Hansen, Laust Hvas Mortensen, Rudi Westendorp.

**Writing – review & editing:** Anne Vinkel Hansen, Laust Hvas Mortensen, Stella Trompet, Rudi Westendorp.

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
