## [Decision Letter · Decision Letter 0]

10 Nov 2020

PONE-D-20-34063

Health Care Expenditure in the last five Years of Life is driven by Morbidity, not Age: A national Study of spending Trajectories in Danish Decedents over Age 65

PLOS ONE

Dear Dr. Hansen,

Thank you for submitting your manuscript to PLOS ONE. After careful consideration, we feel that it has merit but does not fully meet PLOS ONE’s publication criteria as it currently stands. Therefore, we invite you to submit a revised version of the manuscript that addresses the points raised during the review process.

We look forward to receiving your revised manuscript.

Kind regards,

Sreeram V. Ramagopalan

Academic Editor

PLOS ONE

Journal Requirements:

"The authors have declared no competing interests exist. "

We note that one or more of the authors are employed by a commercial company: Data Science Lab.

3.1. Please provide an amended Funding Statement declaring this commercial affiliation, as well as a statement regarding the Role of Funders in your study. If the funding organization did not play a role in the study design, data collection and analysis, decision to publish, or preparation of the manuscript and only provided financial support in the form of authors' salaries and/or research materials, please review your statements relating to the author contributions, and ensure you have specifically and accurately indicated the role(s) that these authors had in your study. You can update author roles in the Author Contributions section of the online submission form.

3.2. Please also provide an updated Competing Interests Statement declaring this commercial affiliation along with any other relevant declarations relating to employment, consultancy, patents, products in development, or marketed products, etc.  

Reviewers' comments:

Reviewer's Responses to Questions

**Comments to the Author**

1. Is the manuscript technically sound, and do the data support the conclusions?

Reviewer #1: Yes

2. Has the statistical analysis been performed appropriately and rigorously? 

Reviewer #1: Yes

3. Have the authors made all data underlying the findings in their manuscript fully available?

Reviewer #1: No

4. Is the manuscript presented in an intelligible fashion and written in standard English?

Reviewer #1: Yes

5. Review Comments to the Author

Reviewer #1: Thank you very much for giving me the opportunity to review the manuscript: PONE-D-20-34063. In this study the authors aimed to identify the most common health care spending trajectories over last five years of life, as well as the determinants of following a given trajectory in Denmark. The authors concluded that spending at the end of life is largely determined by the presence of comorbidities and age, and cause of death only determine the distribution of expenses.

The manuscript is very well-written and is touching on a relevant topic of spending trajectories and determinants of healthcare expenditure toward the end of life. Availability of registry data and universal healthcare system has made it possible to have large sample size and near complete information. I have only a few minor suggestions that are listed below:

• Minor language revision could help the manuscript easier to read. For example: line 17-18, “…...but whether there is any real potential for cost reductions there is still in question,…”

• Line 88: please explain the abbreviation the first time it is used (DRG-grouped National Patient Register)

• Line 80: Please explain briefly why dementia was split as separate category.

• Table 2: What is the difference between “Total health care expenditure over last five years of life” and “Total health care expenditure in last year of life”?

6. PLOS authors have the option to publish the peer review history of their article (what does this mean?). If published, this will include your full peer review and any attached files.

Reviewer #1: No

---

## [Author Response · Author response to Decision Letter 0]

1 Dec 2020

REVIEWER 1: Minor language revision could help the manuscript easier to read. For example: line 17-18, “…...but whether there is any real potential for cost reductions there is still in question,…”

OUR RESPONSE: We have attempted to go over the language for legibility. I really had gone back and forth about that “there” there � but you’re right. 

REVIEWER 1: Line 88: please explain the abbreviation the first time it is used (DRG-grouped National Patient Register)

OUR RESPONSE: Done. 

REVIEWER 1: Line 80: Please explain briefly why dementia was split as separate category.

OUR RESPONSE: We have added the explanation:”… since this disease as a cause of death is of special interest for deaths at higher ages”. 

REVIEWER 1: Table 2: What is the difference between “Total health care expenditure over last five years of life” and “Total health care expenditure in last year of life”?

OUR RESPONSE: The latter figures are for the last twelve months of life, not last five years. We have amended the table text to make this clear.

---

## [Editor Report · Decision Letter 1]

3 Dec 2020

Health Care Expenditure in the last five Years of Life is driven by Morbidity, not Age: A national Study of spending Trajectories in Danish Decedents over Age 65

PONE-D-20-34063R1

Dear Dr. Hansen,

We’re pleased to inform you that your manuscript has been judged scientifically suitable for publication and will be formally accepted for publication once it meets all outstanding technical requirements.

Kind regards,

Sreeram V. Ramagopalan

Academic Editor

PLOS ONE
---

## [Editor Report · Acceptance letter]

9 Dec 2020

PONE-D-20-34063R1 

Health Care Expenditure in the last five Years of Life is driven by Morbidity, not Age: A national Study of spending Trajectories in Danish Decedents over Age 65 

Dear Dr. Hansen:

I'm pleased to inform you that your manuscript has been deemed suitable for publication in PLOS ONE. Congratulations! Your manuscript is now with our production department. 

Kind regards, 

on behalf of

Dr. Sreeram V. Ramagopalan 

Academic Editor

PLOS ONE